# Molecular Mechanisms Involved in Hypoxia-Induced Alterations in Bone Remodeling

**DOI:** 10.3390/ijms23063233

**Published:** 2022-03-17

**Authors:** Ricardo Usategui-Martín, Ricardo Rigual, Marta Ruiz-Mambrilla, José-María Fernández-Gómez, Antonio Dueñas, José Luis Pérez-Castrillón

**Affiliations:** 1Department of Cell Biology, Histology and Pharmacology, Faculty of Medicine, University of Valladolid, 47003 Valladolid, Spain; josefg@med.uva.es; 2IOBA, University of Valladolid, 47011 Valladolid, Spain; 3Department of Biochemistry, Molecular Biology and Physiology, Faculty of Medicine, University of Valladolid, 47003 Valladolid, Spain; rrigual@ibgm.uva.es; 4IBGM, University of Valladolid, 47003 Valladolid, Spain; 5Department of Surgery, Faculty of Medicine, University of Valladolid, 47003 Valladolid, Spain; maria.ruiz@uva.es; 6Department of Medicine, Faculty of Medicine, University of Valladolid, 47003 Valladolid, Spain; antonio.duenas@uva.es; 7Department of Toxicology, Río Hortega University Hospital, 47012 Valladolid, Spain; 8Department of Internal Medicine, Río Hortega University Hospital, 47012 Valladolid, Spain

**Keywords:** hypoxia, bone remodeling, HIF, osteoclast, osteoblast, oxygen

## Abstract

Bone is crucial for the support of muscles and the protection of vital organs, and as a reservoir of calcium and phosphorus. Bone is one of the most metabolically active tissues and is continuously renewed to adapt to the changes required for healthy functioning. To maintain normal cellular and physiological bone functions sufficient oxygen is required, as evidence has shown that hypoxia may influence bone health. In this scenario, this review aimed to analyze the molecular mechanisms involved in hypoxia-induced bone remodeling alterations and their possible clinical consequences. Hypoxia has been associated with reduced bone formation and reduced osteoblast matrix mineralization due to the hypoxia environment inhibiting osteoblast differentiation. A hypoxic environment is involved with increased osteoclastogenesis and increased bone resorptive capacity of the osteoclasts. Clinical studies, although with contradictory results, have shown that hypoxia can modify bone remodeling.

## 1. Introduction

Bone is one of the most metabolically active tissues and is continuously renewed to keep the skeleton healthy. This continuous remodeling allows bone to adapt to the changes required for healthy functioning and maintain bone strength. Bone has three fundamental functions: support for the muscles, the protection of vital organs, and as a reservoir of calcium and phosphorus to carry out various metabolic functions [1]. Osteoblasts, osteoclasts, and osteocytes are the principal cell components of the bone tissue. Bone extracellular matrix has two components, a mineral part comprising hydroxyapatite (70–90%) and an organic part of primarily collagen (approx. 90%) and non-collagenous proteins such as sialoprotein, osteonectin, osteopontin, and osteocalcin (approx. 10%) [1,2,3]. In mature bone, two histological types can be identified: cortical and trabecular bone. Cortical bone is compact, dense, ordered bone and represents 80% of total bone mass. Trabecular bone is lighter and less compact, with an irregular structure [1,4,5].

Sufficient oxygen is required to maintain optimal functioning of cells and tissues. In the absence of sufficient oxygen, under hypoxic conditions, cells must make molecular and physiological adjustments to prolong their survival [6]. In this scenario, evidence has shown that hypoxia may influence bone health [7,8,9]. Thus, this review aimed to examine the molecular mechanisms involved in hypoxia-induced alterations in bone remodeling. We made a comprehensive review of the literature through MEDLINE, PubMed, Web of Science, Scopus, and Embase electronic databases. Potentially relevant articles were sought by using the search terms in combination as Medical Subject Headings (MeSH) terms and text words: “bone”, “bone remodeling”, “hypoxia”, “bone metabolism”, “osteoclast”, “osteoblast” and “osteocyte”. In addition, we scanned the reference lists of the retrieved publications to identify additional relevant articles. The search was supplemented using the MedLine option ‘Related Articles’. No language restrictions were applied. The abstracts for each article were studied to ensure relevance and significance.

## 2. Bone Remodeling and Cells Involved

The purpose of bone remodeling is to maintain bone biomechanical stability and is carried out in an anatomical and functional structure called a bone remodeling unit (BMU). BMU involves four types of cells: bone lining cells, osteocytes, osteoclasts, and osteoblasts. The lining cells, from an osteoblastic lineage, cover the bone surface during the stable phase of remodeling. The most numerous bone cells are the osteocytes, which act mechanistically by detecting areas of weakness in the skeleton and sending signals to activate remodeling. Osteocytes are differentiated osteoblasts encased in the bone matrix. The osteoclasts are multinucleated cells that carry out bone resorption and communicate with the osteoblasts to regulate bone formation [10].

The remodeling cycle is 120 days and involves the removal of mineralized bone by osteoclasts, followed by the formation of bone matrix by osteoblasts that subsequently mineralize. The bone remodeling cycle consists of the following phases: initiation/activation phase of site-specific remodeling; resorption phase, during which osteoclasts digest bone; reversal phase, in which mesenchymal stem cells and osteoblast progenitors are recruited on the bone surface; the formation phase, in which osteoblasts are activated and lay down new bone; and the mineralization phase [11]. In the first three weeks, osteoclast-mediated bone resorption and subsequent osteoclast–osteoblast coupling occur.

The bone remodeling cycle is regulated by systemic and local factors. The principal systemic factors are parathyroid hormone (PTH), growth hormone, calcitriol, thyroid hormones, glucocorticoids, and sex hormones. In addition, regulators such as insulin-like growth factors (IGFs), tumor growth factor-beta (TGF-β), bone morphogenetic proteins (BMP), prostaglandins, and cytokines are also involved. A large number of cytokines and growth factors affect bone cell functions and act as local regulators. The receptor activator of the NF-κB (RANK)-receptor activator of the NF-κB ligand (RANKL)-osteoprotegerin (OPG) pathway also is crucial in bone remodeling regulation [11,12].

### 2.1. Osteocytes

The osteocytes are cells differentiated from osteoblasts that are included within the bone matrix in such a way that approximately 5–20% of osteoblasts are transformed into osteocytes. The precise mechanism and the molecular and genetic regulation by which an osteoblast differentiates into an osteocyte that embeds into the bone matrix are not yet fully understood. It has been hypothesized to be a passive process in which a subpopulation of osteoblasts stop their synthesis of bone matrix and become buried cells beneath the bone matrix synthesized by other osteoblasts. On the other hand, another hypothesis is that it may be an active and invasive process in which is required a matrix degradation for the formation of the osteocyte lacuna [13]. These cells express similar genes to osteoblasts, being particularly important in the expression of genes involved in bone mineralization and phosphorus metabolism. Osteocytes embedded in the mineralized bone matrix are multifunctional cells with many key regulatory roles in bone remodeling and mineral homeostasis. Osteocytes control bone metabolism by regulating osteoclasts and osteoblasts and are mechanosensory cells that regulate and coordinate skeletal responses. Osteocytes play a crucial role in bone calcium deposition and serve as endocrine cells that are important in the control of phosphate homeostasis [13,14].

Osteocytes are the principal producers of sclerostin (encoded by the *SOST* gene). Sclerostin antagonizes several members of BMP and binds to LPR5/LPR6, blocking the canonical Wnt pathway inhibiting the activity of osteoblasts [15]. In addition to blocking bone formation, osteocytes also stimulate the resorption process by secreting RANKL or stimulating RANKL production by osteoblasts or other cells of the osteoblastic lineage. Osteocytes also produce OPG, which competes with RANKL and thus blocks bone resorption [15,16].

### 2.2. Osteoclasts

Osteoclasts have two main functions, bone resorption and the initiation of bone formation through communication with osteoblasts. The osteoclasts are directed to the BMU from the bone marrow or from osteoclast precursors located in the bloodstream [17]. In the osteoclast surface, numerous receptors determine their proliferation, differentiation, and survival. One of the most important is RANK, the binding site of RANKL. The expression of RANK is stimulated by the macrophage colony-stimulator factor (M-CSF) secreted by osteoblasts and bone marrow stromal cells. The RANK–RANKL interaction activates the NF-κB pathway and different kinases, whose final effector is the NAFATc1 factor, which is translocated to the nucleus, activating the expression of genes involved in osteoclast proliferation and maturation. Other important receptors for TNF α, Src, and TERM2 also cooperate in the activation of the NAFATc1 factor [18].

Osteoclast activation is necessary for bone resorption. Firstly, osteoclasts must adhere to the bone surface, carried out by osteoclast podosomes with the participation of integrins, creating the resorption gap. Hydrogen ions are released in the resorption gap, which acidifies the medium and allows the activation of hydrolytic enzymes such as cathepsin, which degrades the bone matrix. The detached material (collagen and calcium) is reabsorbed by osteoclasts and subsequently released to the exterior [17]. Finally, once the process is complete, apoptosis, the programmed cell death of osteoclasts, occurs. However, before this, osteoblast activation must take place. This occurs in three ways. Growth factors embedded in the bone matrix, called matricins, are released due to the activity of the osteoclasts, the most important of which are TGF-β, IGF-1, and BMP2. The osteoblasts are also activated by osteoblast–osteoclast contact through transmembrane proteins such as the ephrin system. Finally, the release of osteoclastokines, which may be stimulatory or inhibitory, is also important [19].

### 2.3. Osteoblasts

Osteoblasts are cell derived from mesenchymal progenitors. Different populations of osteoblasts are distinguished; osteoblast precursors from the resorption gap attracted by substances released from the bone matrix, surface osteoblasts activated by osteocytes, and osteoblasts buried in the bone matrix. Osteoblasts are cells with a basophilic cytoplasm, abundant mitochondria, and a highly-developed Golgi apparatus. This structure explains their ability to synthesize proteins such as alkaline phosphatase, osteocalcin, and type I collagen to form osteoid, which is then mineralized with calcium hydroxyapatite crystals [20,21].

Transcription factors such as Sox9, Runx2, and Atf-4 play an important role in osteoblast differentiation and maturation. The transcription factor Sox9 is not expressed in mature osteoblasts but is crucial in the differentiation from pre-osteoblasts. Runx2 is involved in differentiation and is also important in the function of mature osteoblasts [22]. Atf-4 regulates osteocalcin expression, an osteoblastic protein involved in glucose and RANKL regulation promoting osteoblast differentiation and its function. These transcription factors are activated through exogenous factors such as TGF-β, IGF-1, and fibroblast growth factor (FGF) [19,23]. During resorption, TGF-β is released from the bone matrix, promoting the proliferation of osteoblast precursors and the arrival of mature osteoblasts into the resorption gap. TGF-β also stimulates osteoblast proliferation and increases the synthesis of the matrix. TGF-β inhibits late osteoblast maturation and osteocyte apoptosis, and therefore is crucial in the osteoblast/osteoclast ratio. IGF-1 and FGF develop an anabolic function by activating osteoblast maturation and promoting precursor chemotaxis [19,23,24]. These factors act on receptors located on the surface of the osteoblast, activating metabolic pathways which stimulate the transcription factors previously described. Of particular note are the Notch pathway, the Hedgehog signaling pathway, the Wnt pathway, and the BMP pathway [19,25,26]. In addition to genetic control of these signaling pathways, epigenetic control through micro RNA (miRNA) and small fragments of other non-coding RNA (ncRNA) that regulate post-transcriptional gene expression by inhibiting their translation or stimulating their degradation are important [27].

## 3. Hypoxia: Local Effects on Bone Metabolism Cells

Hypoxia is a condition in which cell function in limited by deprivation of adequate oxygen concentration. The cellular hypoxic response is initiated by hypoxic stimuli such as low oxygen pressure, poor oxygen diffusion-perfusion, among others, and is mediated by hypoxia-inducible factors (HIF). HIF consists of two subunits: an HIF-α subunit (oxygen-sensitive) and a constitutively expressed unit, HIF-β [28,29]. In normoxia, HIF-α is hydroxylated by prolyl hydroxylase domain isoforms (PHDs) and is then polyubiquitinated by the von Hippel-Lindau (VHL) protein for proteasomal degradation. In normoxia, factor-inhibiting hypoxia (FIH) hydroxylates HIF-α, inhibiting its transcriptional capacity [30,31]. In hypoxia, the activity of PHD and FIH are reduced, leading to the accumulation of HIF-α and its translocation to the nucleus. In the nucleus, the heterodimer HIF-α/HIF-β is formed and forms a complex with the cAMP response element-binding (CREB), the binding protein (CBP1), and histone acetyltransferase p300. This complex binds to the hypoxia response elements (HRE) of HIF target genes to activate the expression of more than 200 genes, initiating the stimulation of several cellular pathways aimed at survival in a hypoxic environment [29,32]. The principal pathways activated by HIF-α are angiogenesis, glycolysis, programmed cellular death, and pH regulation to enhance the oxygen in the cells by increasing the concentration of the oxygen transporter (hemoglobin) by EPO or the oxygen flow by activating the sympathetic system [33,34] (Figure 1).

The level of oxygen in bone tissue is reported to be around 6.6–8.5% [35]. Bearing this in mind, it is conceivable that exposure to low concentrations of oxygen can influence bone cellular homeostasis through stimulation of HIF [36,37]. The results described on the influence of hypoxia on the main bone remodeling cells are not uniform and sometimes contradictory. This will depend on the methodology used, as they may vary depending on the degree of hypoxia or the hypoxia/reoxygenation models used. Taking into account the methodology used, generically, it may be that hypoxia inhibits the differentiation and activation of the osteoblasts and induces the activation and activity of osteoclasts. Therefore, the result is an alteration of the bone microarchitecture, facilitating a decrease in strength and an increased risk of bone fragility fracture.

### 3.1. Hypoxia and Osteoblasts

It has been reported that osteoblast activity may decrease in low oxygen environments. In this line, it has been shown that osteoblasts cultured in a 2% oxygen environment decreased their bone formation activity 10-fold. The hypoxic environment delayed the growth and differentiation of osteoblasts [38], and osteoblastogenesis was suppressed in short-term exposure of rats to hypoxia [39]. Figure 2 shows the main events that occur in osteoblasts in hypoxia. Hypoxia could be associated with reduced Runx2 expression, causing a reduction in the differentiation of immature osteoblasts from multipotent mesenchymal cell differentiation [40,41]. The reduced Runx2 could be caused by the expression of Twist that is activated by HIF-1α [42]. In addition, HIF-2α impairs osteoblast differentiation by upregulating Sox9, which inhibits the expression of Runx2 and Sp7, factors involved in osteoblast differentiation [43]. Hypoxia has been associated with inhibition of the phosphatidylinositol 3-kinase (PI3K)/Akt pathway, which plays an important role in anti-apoptotic and survival signals in osteoblasts [44]. Hypoxia could also inhibit osteoblastogenesis, as it may increase induced *SOST* gene expression and therefore increase the expression of sclerostin in the osteocytes [45]. This is a controversial idea; it is also reported that hypoxia decreases sclerostin expression [46]. Hypoxia is also associated with inhibition of osteoblast matrix mineralization. In the matrix, mineralization is crucial in a series of post-translational modifications of collagen carried out by the PHD and lysyl oxidase oxygen-dependent enzymes, and these enzymes are reduced in hypoxia [38,47]. It has been reported that hypoxia reduces the expression and activity of alkaline phosphatase (ALP) [38].

An agonist effect of HIF-1α on bone formation is determined by the osteogenesis–angiogenesis coupling. Hypoxia stimulates increases in vascular endothelial growth factor (VEGF), which promotes osteo-angiogenesis and enhances bone formation. However, it is reported that VEGF may induce an increase in the resorptive activity of osteoclasts [48]. Furthermore, osteoclasts are stimulated by adenosine triphosphate (ATP), which is released by osteoblasts during hypoxia [39].

### 3.2. Local Effects of Hypoxia on Osteoclasts

Osteoclast numbers and activity increase in a hypoxic environment. It is reported that osteoclast activity is increased 21-fold in exposures to 2% of oxygen. In addition, it has also been reported that 2% of oxygen increased resorption pit formation 10-fold [49,50,51]. Figure 3 shows the principal events promoted by hypoxia in the differentiation, maturation, and activity of osteoclasts. It has been suggested that PHD, HIF-1α, and HIF-2α play direct roles in the resorptive capacity of osteoclasts. Hypoxia decreases PHD, promoting the expression of pro-resorptive genes that it might initially inhibit [52]. In addition, HIF-1α activates the expression of pro-resorptive genes and glycolytic activity, stimulating bone resorption [53]. However, the resorptive ability of osteoclasts is increased because HIF-2α increases mineral resorption by activating the expression of genes involved in osteoclast activity (TRAP, CTSK, and NFATC1) [54].

Hypoxia also promotes osteoclastogenesis: HIF-1α and HIF-2α have been associated with increased expression of osteoclast-fusion-related genes, accelerating osteoclast cell fusion [52,54]. Hypoxia has been associated with a suppression of OPG, which favors the RANK–RANKL interaction and the NF-κB pathway [55,56]. TRAF6 is an adapter of RANK, promoting osteoclastogenesis [57], and HIF-2α may upregulate the expression of TRAF6 [54]. Osteoblast–osteoclast crosstalk plays an important role in the response to osteoclasts during hypoxia. VEGF (synthesized by osteoblasts during hypoxia) increases the resorptive activity of osteoclasts [48], which are stimulated by ATP, which is released by osteoblasts in a hypoxic environment [39].

## 4. Hypoxia, Erythropoietin, and Bone Remodeling

During hypoxia, one of the genes targeted by the HIF-α/HIF-β complex is erythropoietin (*EPO*), which encodes a hormone crucial to stimulating red blood cell production. The EPO receptor (EPO-R) is located in erythroid cells in the bone marrow, and EPO binds with EPO-R to activate the STAT3 and STAT5 pathways for erythropoiesis [58].

There are several reports that associate EPO-related diseases and alterations in bone metabolism [59,60]: in this scenario it has been reported that EPO stimulates osteogenesis. Osteogenesis due to EPO could be associated with the expression of EPO-R in osteoblasts and the activation of the mammalian target of the rapamycin (mTOR), JAK2, and PI3K pathways [61,62]. The expression of EPO is associated with angiogenesis activation which, as described above, is crucial in osteoblast–osteoclast crosstalk during hypoxia [63,64]. In addition, it is also reported that EPO could activate the differentiation of osteoblasts through direct interaction with osteoblast precursors or by stimulating BMP production [61,62,65]. However, EPO could be associated with reduced osteoblast mineralization [66,67] through the bone–kidney–parathyroid gland axis [68]. Paradoxically, an association between EPO and increased bone resorption has been reported. Osteoclastogenic activity is also activated by the JAK2 and PI3K signaling pathways [67,69]. EPO has been associated with the differentiation of pre-osteoclasts to mature osteoclasts [70]. The discrepancies between bone formation and bone resorption associated with EPO are poorly understood but may result from large differences in the experimental models used. Nevertheless, it seems clear that EPO plays a role in modulating bone cell responses under hypoxic conditions.

## 5. Vitamin D Metabolism, Inflammation, Hypoxia, and Bone

Vitamin D is crucial in phosphorous and calcium metabolism, cell proliferation, and the control of innate and adaptative immunity. The vitamin D receptor (VDR) is activated by the binding of calcitriol (1-alfa, 25-dihidroxicolecalciferol) to form a complex with the retinoid X receptor (RXR). The calcitriol/VDR/RXR complex migrates to the nucleus to activate the expression of genes involved in vitamin D signaling [71]. In this scenario, it is suggested that hypoxia-associated alterations in bone metabolism may also be mediated by vitamin D/VDR signaling.

In immune cells, it is known that HIF-1α activates the expression of cytokines such as interleukin-1β (IL-1β), IL-6, tumor necrosis factor-α (TNF-α), and interferon-gamma (IFN-γ), promoting an inflammatory response to hypoxia [72,73]. Activation of the transcription of inflammatory-related genes is through NF-κB pathway activation by HIF-1α [74]. In osteoclasts, the NF-κB pathway can also be activated by HIF-α [54]. In non-stimulated cells, the NF-κB pathway is not activated. The p65 component of NF-kB in the cell cytoplasm binds to IkB proteins. After cell stimulation, the IkB proteins are phosphorylated and ubiquitinated to be degraded via the proteasome pathway. This allows the p65 component of NF-κB to be translocated to the nucleus, activating the expression of target genes of the NF-κB pathway [75]. In this scenario, it is reported that calcitriol inhibits p65 translocation to the nucleus and the phosphorylation of the IkB protein [76]. Thus, vitamin D has been associated with a reduction in NF-κB pathway activation in immune cells and osteoclasts [77,78,79]. In addition, in osteoclasts, vitamin D can suppress HIF-1α expression [80] and TNF-α downregulates the expression of VDR [79].

## 6. Hypoxia and Bone Remodeling in Clinical Studies

In humans, there is no good model of the effects of chronic hypoxia on bone metabolism and, in addition, there are other associated factors. In this scenario, contradictory results have been reported. An association between long-term sustained hypoxia exposure and a reduction in several indices of bone health has been reported [7,8,9]. For example, Basu et al. (2013) studied members of the Indian army for four months at altitudes of 5400–6700 m. In contrast, with short exposures, where no alterations were observed [81], this report studied humans at a simulated 4000 m altitude for 21 days. As there is no good model analyzing the effects of chronic hypoxia in clinical studies, one option could be obstructive sleep apnea syndrome, in which nocturnal hypoxia plays a key role. However, it must be taken into account that intermittent hypoxia has different pathophysiological mechanisms. Bone remodeling markers have a diurnal/nocturnal variation. Resorption markers increase during the night, with a peak in the early morning, and decrease in the late afternoon. In the case of the formation marker, it is similar but less intense [82]. Tomiyama et al. (2008) described a link between obstructive sleep apnea and increased bone resorption, and found a positive relationship between apnea/hypopnea episodes and urinary carboxy-terminal collagen crosslinks (CTX) [83]. A meta-analysis of 112,258 subjects reported that sleep apnea syndrome was a risk factor for osteoporosis [84]. Individual studies showed an association between sleep apnea syndrome and the preservation of bone mineral density (BMD) [85,86]. Obstructive sleep apnea has also been associated with the risk of fragility fracture [87]. Analyzing the hypoxic consequences in human studies, it seems clear that hypoxic conditions can modify bone remodeling, with the level of exposure, time, and frequency being crucial. Future research should focus on understanding the molecular response to hypoxia and its consequences in bone remodeling.

## 7. Conclusions

The molecular response to hypoxia is mediated by HIF. In bone, hypoxia has been associated with decreased osteoblast differentiation and activity and increased osteoclast maturation and activity. Hypoxia is associated with reduced bone formation and reduced osteoblast matrix mineralization due to the hypoxia environment inhibiting osteoblast differentiation (mainly mediated by Runx2, Sox9, Wnt, and PI3K/Akt signaling pathways). A hypoxic environment is associated with increased osteoclastogenesis and osteoclast bone resorptive capacity. EPO and vitamin D metabolism could play a key role in modulating the bone molecular response to hypoxia. In the case of clinical studies, contradictory results have been published, although it seems clear that a hypoxic environment can modify bone metabolism. In this scenario, further research is necessary to understand the bone molecular response to hypoxia and its possible clinical consequences on bone metabolism.

## Figures and Tables

**Figure 1 ijms-23-03233-f001:**
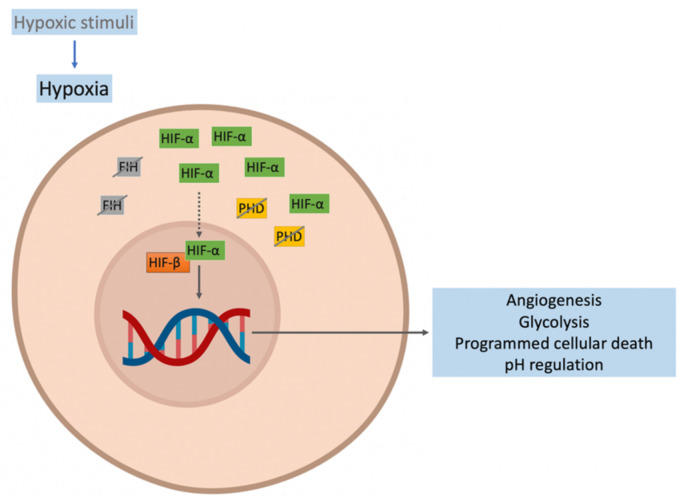
Response to hypoxia. The activity of the prolyl hydroxylase domain (PHD) and factor-inhibiting hypoxia (FIH) are reduced, favoring hypoxia-inducible factor (HIF)-α accumulation and its translocation to the nucleus. In the cell nucleus, the HIF-α/HIF-β heterodimer is formed. This forms a complex that binds to the hypoxia response elements (HRE) of HIF target genes and activates the expression of more than 200 genes, initiating the activation of cellular pathways to enhance the oxygen in the cells.

**Figure 2 ijms-23-03233-f002:**
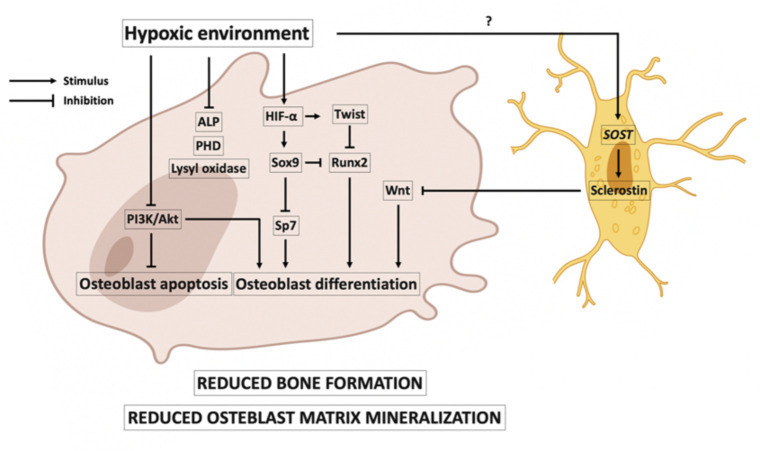
Effects of hypoxia on osteoblasts. A hypoxic environment is associated with a reduction in bone formation and osteoblast matrix mineralization. Hypoxia-inducible factor (HIF)-α is associated with the expression of Twist, which downregulates Runx2 expression and causes a reduction in osteoblast differentiation. Runx2 expression is also reduced by the expression of Sox9, which is promoted by HIF-α. Sox9 factor also reduces the expression of Sp7, another factor involved in osteoblast differentiation. Hypoxia has also been associated with inhibition of the phosphatidylinositol 3-kinase (PI3K)/Akt pathway, favoring osteoblast apoptosis. In a hypoxic environment, in osteocytes, there could be an increase in *SOST* gene expression and therefore an increase in the expression of the sclerostin protein, inhibiting the Wnt pathway and thus, osteoblast differentiation. In hypoxia, reduced osteoblast matrix mineralization is associated with reduced activity of prolyl hydroxylase domain (PHD), lysyl oxidase oxygen, and alkaline phosphatase (ALP).

**Figure 3 ijms-23-03233-f003:**
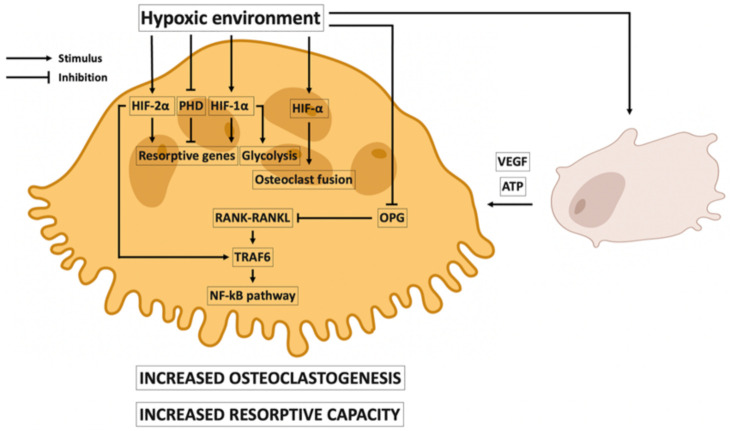
Effects of hypoxia on osteoclasts. Under hypoxic conditions osteoclastogenesis is increased. Hypoxia-inducible factors 1α and 2α (HIF-1α and HIF-2α) activate osteoclast fusion. HIF-2α is also involved in the expression of TRAF6, increasing NF-κB pathway activation and osteoclast maturation. NF-κB is also activated because hypoxia decreases osteoprotegerin (OPG) expression. Hypoxia is associated with increased osteoclast resorptive capacity. Pro-resorptive gene expression is activated by HIF-1α and HIF-2α and by the inhibition of the prolyl hydroxylase domain (PHD). HIF-1α is also involved in the stimulation of glycolytic activity. In the osteoclast response to hypoxia, osteoblast–osteoclast crosstalk is crucial. Resorptive osteoclast activity is stimulated by vascular endothelial growth factor (VEGF) and adenosine triphosphate (ATP); VEGF and ATP are released by osteoblasts in a hypoxic environment.

## Data Availability

Not applicable.

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
