# Peer review of "Molecular Mechanisms Involved in Hypoxia-Induced Alterations in Bone Remodeling"

_ijms, 2022, doi:10.3390/ijms23063233_

Round 1

Reviewer 1 Report

The article is a well organized analysis of the effects of hypoxia on all the components of bone,starting with the description of each component  followed by the effect of hypoxia on the component and presenting all the papers regarding the topic.This organization of the revew results very useful to students  as well to reserchers of the field for the exhaustive literature reported.

Author Response

March, 11th 2022

Dear Sir:

Enclosed please find our revised manuscript by Ricardo Usategui-Martín, Ricardo Rigual, Marta Ruiz-Mambrilla, José-María Fernández-Gómez, Antonio Dueñas and José Luis Pérez-Castrillón, for consideration for publication in the INTERNATIONAL JOURNAL OF MOLECULAR SCIENCES. The authors greatly appreciate the valuable critiques and comments of the Editor and Reviewers and the opportunity to revise our manuscript. We believe that the revisions based on their recommendations have improved the value and accuracy of our manuscript and we hope that you may consider it suitable for publication in your journal. Following the recommendations, the manuscript has been revised by a Scientific English Editor. As requested, we are sending a plain manuscript with accepted changes and a manuscript with tracked changes. The point-by-point answers to editor and reviewers’ comments are provided below.

Yours sincerely,

Prof. José Luis Pérez-Castrillon and Dr. Ricardo Usategui Martín on behalf of the authors

School of Medicine. University of Valladolid. Valladolid. Spain

e-mail: uvacastrv@gmail.com and rusategui@gmail.com

Reviewer 1 comments

The article is a well organized analysis of the effects of hypoxia on all the components of bone, starting with the description of each component  followed by the effect of hypoxia on the component and presenting all the papers regarding the topic. This organization of the review results very useful to students  as well to reserchers of the field for the exhaustive literature reported.

We are grateful for this comment.

Reviewer 2 Report

  • The authors explain the differences between organic and inorganic phases of bone matrix. However, this is not correct. The paragraph needs to be rewritten. As described in Carvalho et al. 2018, bone extracellular  matrix has two components: a mineral part comprising hydroxyapatite (70–90%) and an organic part (10–30%) of primarily collagen (approx. 90% of organic matrix) with the rest being non-collagenous proteins (~10%) .
    Carvalho MS, Poundarik AA, Cabral JMS, da Silva CL, Vashishth D. Biomimetic matrices for rapidly forming mineralized bone tissue based on stem cell-mediated osteogenesis. Sci Rep. 2018;8(1):14388.
  • Please rephrase this sentence: “The osteocytes derive from osteoblasts buried in the one matrix.”
  • Please explain better why some osteoblasts transform to osteocytes.
  • Please remove the repeated phrase: “transcription factors such as Sox9, Runx2, or Atf-4”
  • Please correct “TFG-beta”
  • Please rephrase the following paragraph: “Hypoxia may be defined as the point at which cellular function is limited by the concentration of oxygen.”
  • Please provide references for the following sentence: “The level of oxygen in bone tissue is reported to be around 6.6-8.5%”
  • Please rephrase the following paragraph: “Osteoblast activity may be decreased in low oxygen environments: it is reported that bone formation decreased 10-fold with osteoblast cultured to 2% of oxygen.”

Author Response

March, 11th 2022

Dear Sir:

Enclosed please find our revised manuscript by Ricardo Usategui-Martín, Ricardo Rigual, Marta Ruiz-Mambrilla, José-María Fernández-Gómez, Antonio Dueñas and José Luis Pérez-Castrillón, for consideration for publication in the INTERNATIONAL JOURNAL OF MOLECULAR SCIENCES. The authors greatly appreciate the valuable critiques and comments of the Editor and Reviewers and the opportunity to revise our manuscript. We believe that the revisions based on their recommendations have improved the value and accuracy of our manuscript and we hope that you may consider it suitable for publication in your journal. Following the recommendations, the manuscript has been revised by a Scientific English Editor. As requested, we are sending a plain manuscript with accepted changes and a manuscript with tracked changes. The point-by-point answers to editor and reviewers’ comments are provided below.

Yours sincerely,

Prof. José Luis Pérez-Castrillon and Dr. Ricardo Usategui Martín on behalf of the authors

School of Medicine. University of Valladolid. Valladolid. Spain

e-mail: uvacastrv@gmail.com and rusategui@gmail.com

Reviewer 2 comments

The authors explain the differences between organic and inorganic phases of bone matrix. However, this is not correct. The paragraph needs to be rewritten. As described in Carvalho et al. 2018, bone extracellular matrix has two components: a mineral part comprising hydroxyapatite (70–90%) and an organic part (10–30%) of primarily collagen (approx. 90% of organic matrix) with the rest being non-collagenous proteins (~10%). Carvalho MS, Poundarik AA, Cabral JMS, da Silva CL, Vashishth D. Biomimetic matrices for rapidly forming mineralized bone tissue based on stem cell-mediated osteogenesis. Sci Rep. 2018;8(1):14388.

We thank the reviewer for raising our attention to this point, we have modified it.

Please rephrase this sentence: “The osteocytes derive from osteoblasts buried in the one matrix.”

We have rephased it, we have now stated: Osteocytes are differentiated osteoblasts encased in the bone matrix”

Please explain better why some osteoblasts transform to osteocytes.

We have applied the explication in the Osteocytes section.

Please remove the repeated phrase: “transcription factors such as Sox9, Runx2, or Atf-4”

We have removed it

Please correct “TFG-beta”

We have corrected it.

Please rephrase the following paragraph: “Hypoxia may be defined as the point at which cellular function is limited by the concentration of oxygen.”

We have rephased it, we have now stated: “Hypoxia is a condition in which cell function in limited by deprived of adequate oxygen concentration”

Please provide references for the following sentence: “The level of oxygen in bone tissue is reported to be around 6.6-8.5%”

The references is: “Harrison, J. S., Rameshwar, P., Chang, V. V., Bandari, P., & Persis, B. (2002). Oxygen saturation in the bone marrow of healthy volunteers. Blood, 99(1), 394”. We have included it.

Please rephrase the following paragraph: “Osteoblast activity may be decreased in low oxygen environments: it is reported that bone formation decreased 10-fold with osteoblast cultured to 2% of oxygen.”

We have rephased it, we have now stated: “It has been reported that osteoblast activity may decrease in low oxygen environments. In this line, it has been shown that osteoblasts cultured in a 2% oxygen environment decreased their bone formation activity 10-fold

Round 2

Reviewer 2 Report

The authors have addressed the reviewer's comments.